# Therapeutic Efficacy of *Weissella cibaria* CMU and CMS1 on Allergic Inflammation Exacerbated by Diesel Exhaust Particulate Matter in a Murine Asthma Model

**DOI:** 10.3390/medicina58091310

**Published:** 2022-09-19

**Authors:** Kyung-Hyo Do, Kwangwon Seo, Sanggu Kim, Soochong Kim, Geun-Yeong Park, Mi-Sun Kang, Wan-Kyu Lee

**Affiliations:** 1Laboratory of Veterinary Bacteriology and Infectious Diseases, College of Veterinary Medicine, Chungbuk National University, Cheongju 28644, Korea; 2College of Veterinary Medicine, Chungbuk National University, Cheongju 28644, Korea; 3Laboratory of Veterinary Pathology and Platelet Signaling, College of Veterinary Medicine, Chungbuk National University, Cheongju 28644, Korea; 4R&D Center, OraPharm, Inc., Seoul 04782, Korea

**Keywords:** *Weissella cibaria* CMU, *Weissella cibaria* CMS1, asthma, inflammation, bronchial hyper-responsiveness

## Abstract

*Background and Objectives*: Diesel exhaust particulate matter (DEPM) is an air pollutant that is associated with asthma. In this study, the therapeutic efficacy of *Weissella cibaria* strains CMU (Chonnam Medical University) and CMS (Chonnam Medical School) 1, together with the drug Synatura, an anti-tussive expectorant, was investigated in a murine asthma model exacerbated by DEPM. *Materials and Methods*: BALB/c mice were sensitized with ovalbumin (OVA) before intranasal challenge with OVA and DEPM. *W. cibaria* CMU, CMS1, and Synatura were administered orally for 21 days. *Results*: Neither Synatura nor *W. cibaria* strains affected spleen, liver, or lung weights. *W. cibaria* strains CMU and CMS1 significantly reduced the levels of interleukin (IL)-4, OVA-specific immunoglobulin E (IgE), and total lung collagen in bronchoalveolar lavage fluid (BALF), similar to those with Synatura, regardless of the oral dose concentration (*p* < 0.05). In addition, the *W. cibaria* CMU strain significantly alleviated IL-1β, IL-6, IL-12, monocyte chemotactic protein-1, and tumor necrosis factor-α in BALF, whereas the CMS1 strain significantly alleviated IL-10 and IL-12 in BALF (*p* < 0.05); however, Synatura did not show any statistical efficacy against them (*p* > 0.05). All concentrations of *W. cibaria* CMU and low concentrations of *W. cibaria* CMS1 significantly reduced lung bronchiolar changes and inflammatory cell infiltration. *Conclusions*: In conclusion, *W. cibaria* CMU in asthmatic mice showed better efficacy than *W. cibaria* CMS1 in improving asthma exacerbated by DEPM exposure, as well as better results than pharmaceuticals.

## 1. Introduction

Asthma is a chronic disorder of the airway characterized by airway inflammation, a complex interaction of airflow obstruction, and bronchial hyper-responsiveness. Allergic asthma is characterized by goblet cell hyperplasia, subepithelial fibrosis, and smooth muscle hypertrophy [1]. Allergens, air pollution, cigarette smoke, ozone, and other environmental factors can increase the risk of acquiring respiratory disorders [2,3].

Air pollution has been shown to negatively impact human cardiopulmonary health. Diesel exhaust particulate matter (DEPM) is a major component of particulate matter (PM). DEPM consists of gaseous and aerosol phases that include a variety of hazardous chemicals. In June 2012, DEPM was classified as a group 1 human carcinogen. Metals such as zinc and copper, as well as polycyclic aromatic hydrocarbons such as anthracene, naphthalene, and benzo(a)pyrene, can be found on the surfaces of these fine particles [4]. DEPM increases chemokine production in inflammatory diseases such as asthma (Jin et al., 2020). DEPM stimulates T helper (Th) type 2 cells and the secretion of Th-2 cytokines, including interleukin (IL)-4, IL-6, IL-10, and IL-13, and increases immunoglobulin (Ig) E levels [5]. These cytokines can result in airway inflammation, mucus production, bronchial smooth muscle contractions, and the activation of neutrophils, eosinophils, and macrophages [6].

Since their debut in the market, inhaled corticosteroids have been the cornerstone of asthma treatment. Despite the efficacy of inhaled corticosteroids, certain patients may experience drug-related side effects such as dysphonia and thrush [7]. Therefore, new options are being explored, and one of them may be the use of probiotics to prevent asthma [8]. Probiotics have been proven to have immunomodulatory properties and play a role in the pathophysiology of various diseases [9,10]. Lacerda et al., 2021 reported that lactic acid bacteria (LAB) reduced the overall burden of airway allergic inflammation in murine models of asthma [11]. The beneficial effects of the oral administration of probiotics, including *Lactobacillus* (*L.*) *gasseri*, *L. fermentum*, *L. casei*, *L. rhamnosus*, *L. salivarius*, and *Bifidobacterium lactis,* in murine models of asthma have been reported in many studies [12,13,14,15,16]. Furthermore, Carvalho et al., 2020 reported that *L. rhamnosus* improved the lung inflammatory response associated with cigarette smoke by modulating the balance between pro- and anti-inflammatory cytokines [17].

In a previous study, we verified the safety of *Weissella cibaria* strains CMU (Chonnam Medical University) and CMS (Chonnam Medical School) 1 and demonstrated their anti-inflammatory effects in vitro [18,19]. In addition, *W. cibaria* strains CMU and CMS1 have shown anti-microbial and anti-biofilm activities against upper-respiratory-tract-infection-causing pathogens [9]. Therefore, in this study, we investigated the therapeutic efficacy of *W. cibaria* CMU and CMS1 in inflammatory and asthmatic responses using a murine asthma model exacerbated by DEPM.

## 2. Materials and Methods

### 2.1. Reagents and Chemicals for Enzyme-Linked Immunosorbent Assay

Enzyme-linked immunosorbent assay (ELISA) kits for IL-1β, IL-4, IL-6, IL-10, IL-12, IL-13, monocyte chemotactic protein (MCP)-1, tumor necrosis factor (TNF)-α, interferon (IFN)-γ, and ovalbumin (OVA)-specific immunoglobulin E (IgE) were purchased from Invitrogen (Waltham, MA, USA). For measuring the level of total lung collagen, the Sircol Collagen Assay kit (Biocolor Ltd., Belfast, Northern Ireland) was used. NIST^®^ SRM^®^ 2975 (National Institute of Standard and Technology Standard Reference Material) was purchased from Sigma-Aldrich (St. Louis, MO, USA). All kits for ELISA and total lung collagen level measurement were used according to the protocol provided by the supplier. All chemicals used were of the highest commercially available quality.

### 2.2. Preparation of DEPM Suspensions

SRM 2975 is a DEPM collected from a diesel-powered industrial forklift. After preparing the DEPM suspension solution (1 mg/mL) in sterile phosphate-buffered saline (PBS), the DEPM solution was sonicated for 2 min using a sonicator (Sibata; OGAWA SEIKI, Tokyo, Japan) to minimize the aggregation of PM.

### 2.3. Preparation of W. cibaria

*W. cibaria* strains CMU and CMS1 were provided by OraPharm, Inc. (Seoul, Korea). The strains were cultured in De Man, Rogosa, and Sharpe (MRS) broth and agar medium (Becton-Dickinson, BD, USA) at 37 °C for 24 h. After the centrifugation (8000× *g* for 20 min at 4 °C) of the bacterial cells, the pellet was washed twice with sterilized PBS. To adjust the appropriate density for the experiment, live *W. cibaria* CMU and CMS1 were quantified using the plate count method on MRS agar. 

### 2.4. DEPM-Exacerbated Murine Asthma Model and Probiotic Treatment

The mice were given a week to acclimate before the experiments. Five-week-old male BALB/c mice (20 ± 2 g) were housed in a room with controlled temperature (21 ± 2 °C) and humidity (50 ± 5%) under a 12 h light/dark cycle with free access to food and water. All animal experiments were approved by the Institutional Animal Care and Use Committee of Chungbuk National University (approval no. “CBNUA-1526-21-01”; approval date. “1 April 2020”). Intranasal treatment with the DEPM solution was followed by a conventional 21-day, OVA-sensitized murine asthma model. Seven days after acclimatization, eight mice per group were divided as follows: negative control (NC), positive control (PC), Synatura, CMS1-low, CMS1-mid, CMS1-high, CMU-low, CMU-mid, and CMU-high groups. For sensitization, 200 μL of aluminum hydroxide (Al(OH)_3_) saline (1:1) containing 100 μg of OVA (Grade V, Sigma-Aldrich) was injected intraperitoneally to all mice except for the NC group on days 0 and 12 (saline was injected intraperitoneally to the NC group). On days 19 and 20, all groups, except the NC group, were challenged with OVA (25 μg/20 μL per mouse) by intranasal instillation (the NC group was challenged with 20 μL of saline). After the last OVA challenge, 20 μL of DEPM solution (200 μg/20 μL per mouse) was challenged via intranasal instillation in all groups three times over a 3 h interval. The NC group was challenged three times, over a 3 h interval, with 20 μL of PBS via intranasal instillation. Mice were fed either *W. cibaria* CMS1 (CMS1-low: 2 × 10^7^, CMS1-mid: 2 × 10^8^, and CMS1-high: 2 × 10^9^ colony forming unit (CFU)/mouse), *W. cibaria* CMU (CMU-low: 2 × 10^7^ CFU/mouse, CMU-mid: 2 × 10^8^ CFU/mouse, and CMU-high: 2 × 10^9^ CFU/mouse), and Synatura (200 mg/kg; Ahn-Gook Pharmaceuticals Co., Seoul, Korea) or the same volume of PBS (NC and PC) by oral administration from day 0 to day 20. Synatura was used as the active control. The experimental design is illustrated in Figure 1.

### 2.5. Bronchoalveolar Lavage Fluid Collection

After the anesthesia, mice were euthanized, and bronchoalveolar lavage fluid (BALF) was immediately collected. An amount of 1.5 mL of PBS was immediately flushed into the lungs via the trachea to collect the BALF. The collected BALF was immediately placed on ice and centrifuged at 400× *g* for 5 min at 4 °C to collect the supernatants for measuring cytokine levels.

### 2.6. Enzyme-Linked Immunosorbent Assay (ELISA)

To measure the levels of IL-1β, IL-4, IL-6, IL-10, IL-12, IL-13, MCP-1, TNF-α, IFN-γ, and OVA-specific IgE in BALF, ELISA kits (Invitrogen) were used, according to the assay protocol provided by the supplier. All samples and standards were tested in triplicate. Absorbance was measured at 450 nm using a Tecan Sunrise ELISA reader (Männedorf, Switzerland). The concentration of each cytokine was calculated from the standard curve of each cytokine standard.

### 2.7. Histological Examination of Lung Tissue

After the excision of the left lung lobes, the tissues were fixed in 10% neutral-buffered formalin for 24 h. Fixed tissues were sectioned at a thickness of 4 μm after dehydration and paraffin embedding. Tissue sections were then stained with hematoxylin and eosin (H&E) for histological analysis and periodic acid–Schiff (PAS) staining for the histochemical analysis of mucus production. Histological images were obtained using a Slideview VS200 Scanner (Olympus, Tokyo, Japan), and the pulmonary lesions were analyzed based on the scoring criteria on a scale ranging from 0 to 3 (Table 1, Appendix A).

### 2.8. Statistical Analysis

Significant differences (*p* < 0.05) between groups were tested by analysis of variance (ANOVA) and compared by Duncan’s post hoc tests using the software SPSS Statistics version 21.0 (IBM Corp., Armonk, NY, USA).

## 3. Results

### 3.1. Effects of W. cibaria CMU and CMS1 on the Weight of Spleen, Liver, and Lungs in a DEPM-Exacerbated Murine Asthma Model

OVA plus DEPM treatment did not have a significant effect on the weights of the spleen, liver, and lungs in the mice (Figure 2). The weight of the spleen showed a tendency to increase when OVA and DEPM were administered; however, there was no statistically significant difference compared with the negative control group. Synatura-treated and *W. cibaria* CMU-low and high-concentration-treated groups showed an increase compared to the negative control group (*p* < 0.05); however, there was no statistically significant difference from the positive control group. There were no significant differences in liver and lung weights between the groups.

### 3.2. Improvement Effects of W. cibaria CMU and CMS1 on DEPM-Induced Exacerbated Inflammatory Cytokines and Chemokines in BALF

The levels of inflammatory cytokines, chemokines, and total lung collagen are shown in Figure 3. The CMU-high group (139.64 ± 124.4 pg/mL) showed significantly reduced IL-1β levels compared to the PC group (417.00 ± 215.2 pg/mL) (*p* < 0.05). Additionally, the CMU-low (284.50 ± 143.1 pg/mL) and CMU-mid (372.21 ± 257.4 pg/mL) groups showed a tendency to decrease IL-1β levels compared to the PC group. All groups in which the mice were fed *W. cibaria* CMS1 (CMS1-low, 22.44 ± 6.1 pg/mL; CMS1-mid, 29.67 ± 6.4 pg/mL; and CMS1-high, 23.88 ± 4.4 pg/mL) and *W. cibaria* CMU (CMU-low, 24.11 ± 4.3 pg/mL; CMU-mid, 25.15 ± 5.9 pg/mL; and CMU-high, 21.35 ± 4.5 pg/mL) had significantly decreased IL-4 levels compared to the PC group (49.44 ± 12.0 pg/mL) (*p* < 0.05). In the case of IL-10, the CMS1-low (230.75 ±91.4 pg/mL) and CMS1-high (154.04 ± 61.7 pg/mL) groups had significantly lower levels than the PC group (504.82 ± 296.2 pg/mL) (*p* < 0.05). With respect to the level of IL-12, the CMS1-low (248.43 ± 169.6 pg/mL), CMS1-mid (269.07 ± 65.4 pg/mL), CMU-low (180.36 ± 112.7 pg/mL), and CMU-mid (230.20 ± 109.7 pg/mL) groups showed a significant asthma improvement efficacy (*p* < 0.05). All the CMU and CMS1 groups showed significantly decreased OVA-specific IgE and total lung collagen levels (*p* < 0.05). Interestingly, only the CMU-low group showed improved asthma efficacy for MCP-1. The MCP-1 level in the CMU-low group was 269.93 ± 95.3 pg/mL, which was significantly lower than the PC group (409.79 ± 124.2 pg/mL) (*p* < 0.05). For IL-6 and TNF-α, there was a statistically significant decrease in the CMU-high group (IL-6, 24.53 ± 8.7 pg/mL vs. 43.47 ± 31.3 pg/mL; TNF-α, 119.31 ± 55.6 pg/mL vs. 214.94 ± 97.6 pg/mL) (*p* < 0.05) compared to the PC group.

### 3.3. Effects of W. cibaria CMU and CMS1 on Histopathological Changes Exacerbated by DEPM

Histopathological changes and mucosal hyperplasia in the lung tissues were determined using H&E (Figure 4) and PAS staining (Figure 5). Compared to the NC group, moderate to severe bronchiolar changes, including mucosal hyperplasia (goblet cell hyperplasia), subepithelial fibrosis, and epithelial detachment, were observed in the PC group. Lymphocytic infiltrates in the submucosa and smooth muscle area were also observed. In addition, the increased infiltration of inflammatory cells, including lymphocytes, macrophages, and occasional eosinophils, was observed in the peribronchiolar and perivascular lesions. However, bronchiolar changes and inflammatory cell infiltration were significantly alleviated in the CMS1-low, CMU-low, CMU-mid, and CMU-high groups. These improved pathological lesions were comparable to those observed in the Synatura group. Pathological changes intensified with increasing doses of CMS in the CMS-mid and CMS-high groups compared to the CMS1-low group. Consistent with the histopathological lesions, we found an increase in mucus production with the hyperplasia of goblet cells in the PC group compared with that in the NC group via PAS staining. However, mucus production and goblet cell hyperplasia decreased in the Synatura, CMS1-low, CMU-low, CMU-mid, and CMU-high groups.

### 3.4. Alleviating Effects of W. cibaria CMU and CMS1 on Bronchiolar and Inflammatory Changes Exacerbated by DEPM

The results of bronchiolar changes are shown in Figure 6. The *W. cibaria* CMU group scored significantly lower than the PC group, regardless of the dose concentration (*p* < 0.05), whereas in the *W. cibaria*-CMS1-administered group, the CMS1-low and CMS1-mid groups showed asthma improvement (*p* < 0.05). No effects were observed in the CMS1-high group. Inflammatory cell infiltration evaluation also showed that, similar to bronchiolar changes, the CMU-administered group showed significantly lower scores compared to the PC group, regardless of the dose concentration, and showed asthma improvement efficacy (*p* < 0.05). In the CMS1-administered group, only the CMS1-low group showed an improvement in asthma (*p* < 0.05) (Figure 6).

## 4. Discussion

Asthma is one of the most common chronic inflammatory diseases worldwide [7]. The efficacy of organically produced anti-asthmatic medicines in treating severe allergic asthma has recently come to light [20]. Accordingly, probiotics have been proposed as novel asthma treatments and alternative medications. Probiotic LAB, such as *Lactobacillus* and *Bifidobacterium* genera, have the potential to improve allergic diseases by promoting T regulatory cell development and rebalancing the immune response toward a Th1-dominant state [1]. Therefore, probiotics are proposed as new alternatives for the treatment of allergic asthma.

Recently, studies on the probiotic properties of various *W. cibaria* strains have been conducted. *W. cibaria* JW15 has been reported to exhibit anti-microbial activity and immunomodulatory effects, such as increasing nuclear factor (NF)-κB, IL-1β, and TNF-α levels in macrophages [21,22]. Huang et al., 2020 reported that *W. cibaria* MW01, as a potential probiotic, could attenuate the lipopolysaccharide-induced dysfunction of the intestinal epithelial barrier in a Caco-2 cell monolayer model [23]. *W. cibaria* CMU and CMS1 were the first commercialized strains isolated from the saliva of children [24]. It has been confirmed that both strains do not cause hemolytic activities and gelatin liquefaction and are safe strains without antibiotic resistance gene transfer ability and toxin genes [18]. The safety of these strains has also been proven in animal toxicity studies and human clinical trials [25,26]. *W. cibaria* CMU has been reported to exert immunomodulatory effects on IL-6 and IL-8 responses in oral epithelial cells activated by *Fusobacterium nucleatum* [27]. *W. cibaria* CMU has also been reported as an oral probiotic strain, and it exerts anti-inflammatory effects by inhibiting NF-κB activation induced by *Aggregatibacter actinomycetemcomitans* in macrophages [19]. It was also reported that live *W. cibaria* CMU had no cytotoxic effects on macrophages and downregulated the mRNA expression of pro-inflammatory cytokines [19].

Exposure to ambient fine PM has been associated with increased hospitalizations and mortality in patients with chronic obstructive pulmonary disease and asthma [1,2]. DEPM is a major component of airborne PM produced by the incomplete combustion of diesel fuel, and its role in aberrant immune responses to allergic diseases such as asthma has been studied [1]. DEPM is known to increase chemokine production in inflammatory diseases such as asthma [4] and increase levels of IgE and Th2-type cytokines [4]. A previous study showed that LAB ameliorated DEPM-exacerbated allergic inflammation in a murine asthma model [1]. In this study, we evaluated the effects of *W. cibaria* CMU and CMS1, which were orally administered to a murine asthma model exacerbated by DEPM. Interestingly, in our study, we found that OVA and DEPM challenges had no significant effect on the liver, spleen, or lungs. This result was inconsistent with the findings of Jin et al., 2020 that the liver weight of mice in the OVA plus DEPM group did not change, but the spleen and lung weights increased [1]. To evaluate the effectiveness of probiotics in OVA- and DEPM-challenged models of asthma, histopathological changes should be evaluated alongside the classic OVA-induced asthma model.

Cytokines play an important role in regulating the inflammatory responses in asthma [9]. Many Th2-type cytokines and chemokines are involved in the pathophysiology of asthma, including the promotion of airway eosinophilia, bronchial hyper-responsiveness, and an increase in IgE levels [7]. Steinke and Borish (2001) reported that IL-4 is an upstream cytokine that regulates allergic inflammation by promoting Th2 cell differentiation and IgE synthesis [28]. Although IL-12 exhibits a protective function in models of intracellular microbial infection by driving a Th1 response [29], it is notorious as a cytokine that promotes various inflammatory responses associated with autoimmune diseases [30].

This study showed that the oral administration of *W. cibaria* CMU and CMS1 suppressed IL-1β, IL-4, IL-6, IL-10, IL-12, TNF-α, MCP-1, and OVA-specific IgE levels in the BALF. Additionally, both *W. cibaria* CMU and CMS1 significantly reduced IL-4, OVA-specific IgE, and total lung collagen levels. Surprisingly, *W. cibaria* strains were demonstrated to be more effective than Synatura in decreasing the levels of most Th2 cytokines.

Synatura^®^ (AG NPP709), developed by Ahn-Gook Pharmaceuticals Co. (Seoul, Korea) for the prevention and treatment of asthma, was used in the active control group in this study. It is a combined herbal medicine extract containing berberine and hederacoside C, which are active ingredients extracted from *Coptis chinensis* Franch (Ranunculaceae) rhizome and *Hedera helix* L. (ivy; Lamiaceae) leaves, which are anti-inflammatory herbal medicines. Synatura^®^ is a natural new drug that exerts various effects such as anti-tussive, expectorant, anti-inflammatory, anti-histamine, the inhibition of bronchial contraction, and airway remodeling [31]. In this study, the Synatura group exhibited significantly reduced levels of IL-4, OVA-specific IgE, and total lung collagen. *W. cibaria*-CMU- and *W. cibaria*-CMS1-administered group showed significant improvements in levels of IL-1β, IL-4, IL-6, IL-12, MCP-1, TNF-α, OVA-specific IgE, total lung collagen, and IL-4, IL-10, IL-12, OVA-specific IgE, and total lung collagen, respectively.

Recently, attention has been paid to airway remodeling due to cellular and histological changes in airway structure over time [7], for example, the hyperplasia and hypertrophy of goblet cells, the hyperplasia of submucosal glands, the thickening of the epithelial basement membrane and subepithelial fibrosis, the hyperplasia of airway smooth muscle cells, and hypertrophy. Therefore, airway remodeling may cause irreversible airflow limitation, increase airway hyper-responsiveness, and worsen the severity of asthma. Airway remodeling can be induced by cytokines and mediators produced in the chronic allergic inflammatory response of the airways [7].

In the histopathological observation of lung tissue using PAS staining, in the *W. cibaria*-CMU-treated group in our study, the efficacy of inhibiting mucosal hyperplasia was equivalent to that of Synatura. Similar to the PAS staining results, in the evaluation of bronchial changes and inflammatory cell infiltration, the *W. cibaria* CMU group showed significantly lower scores than the PC group, regardless of the dose concentration, whereas the *W. cibaria* CMS1 group showed a significant effect only at a low concentration. Therefore, it was confirmed that *W. cibaria* CMU alleviated asthma better than CMS1 did.

The secretion of chemokines was found to correlate with the severity of airway hypersensitivity and bronchial inflammatory response in asthmatic patients [7]. Among these chemokines, MCP-1 directly induces airway hyper-responsiveness by activating mast cells and releasing leukotriene C4 into the airway. Therefore, it is thought to play a very important role in asthma response [32]. Gonzalo et al. [33] reported that the neutralization of MCP-1 reduced bronchial hyper-responsiveness and decreased the airway inflammatory response in an animal model. MCP-1 is also known to induce the differentiation of undifferentiated T cells into Th2 cells that produce IL-4 [32]. Another study showed that MCP-1 expression was increased in the bronchial epithelial cells of asthmatic patients, and the concentration of MCP-1 in sputum during an acute asthma attack in asthmatic patients was significantly increased compared to that in the asymptomatic phase of the same patient [33]. It has also been reported that the concentration of MCP-1 in the BALF of asthmatic patients with atopic dermatitis was higher than that in the control group.

## 5. Conclusions

In this study, the oral administration of *W. cibaria* CMU and *W. cibaria* CMS1 significantly improved histopathological changes in the lung tissue. In conclusion, the oral administration of *W. cibaria* strains CMU and CMS1 in a murine asthma model showed significant therapeutic efficacy in improving lung inflammation exacerbated by DEPM administration. Specifically, *W. cibaria* CMU showed therapeutic efficacy against allergic asthma.

## Figures and Tables

**Figure 1 medicina-58-01310-f001:**
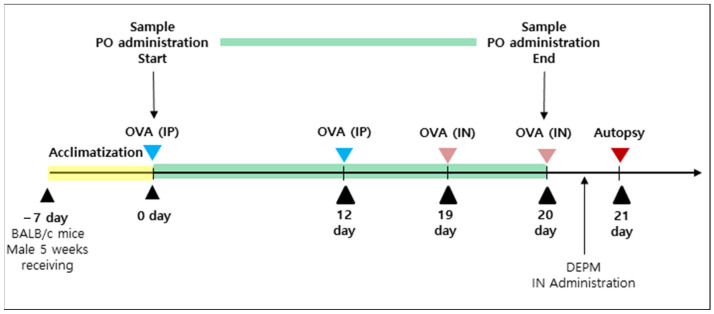
Experimental design to analyze the effects of *Weissella cibaria* CMU (Chonnam Medical University) and CMS (Chonnam Medical Science) 1 on BALB/c mice model of asthma induced by ovalbumin (OVA) and diesel exhaust particulate matter (DEPM). *W. cibaria* CMU and CMS1 were administered per oral (PO) from days 0 to 20. On days 0 and 12, OVA was used to sensitize the mice via intra peritoneal injection (IP) (Blue triangles). On days 19 and 20, mice were challenged with OVA via nasal instillation (IN) (Pink triangles). After the last challenge of OVA, DEPM solution was treated three times over a 3 h interval. On day 21, mice were euthanized, and tissue and blood were collected (Red triangle). Collected tissues were washed with sterilized phosphate-buffered saline. Liver, spleen, and lungs were weighted to analyze the therapeutic efficacy of *W. cibaria* CMU and CMS1. Synatura was used as an active control group.

**Figure 2 medicina-58-01310-f002:**
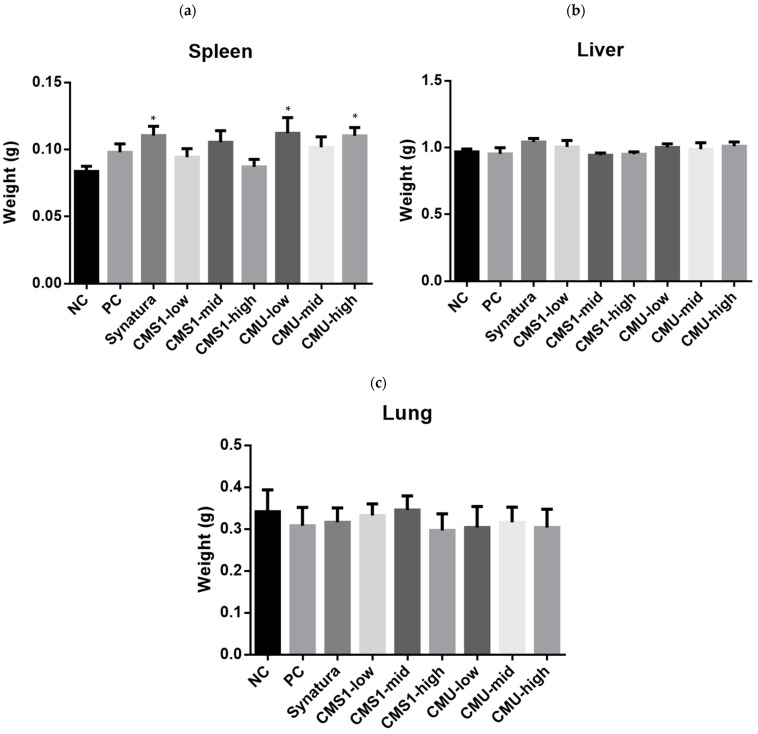
Effects of *W. cibaria* CMU and CMS1 on the weights of the spleen (**a**), liver (**b**), and lungs (**c**) of mice challenged with ovalbumin (OVA) plus diesel exhaust particulate matter (DEPM). Collected tissues were washed with phosphate-buffered saline. Synatura was used as an active control. NC, negative control; PC, positive control. Data represent the mean ± standard deviation. * indicate the statistical differences with negative control (*p* < 0.05).

**Figure 3 medicina-58-01310-f003:**
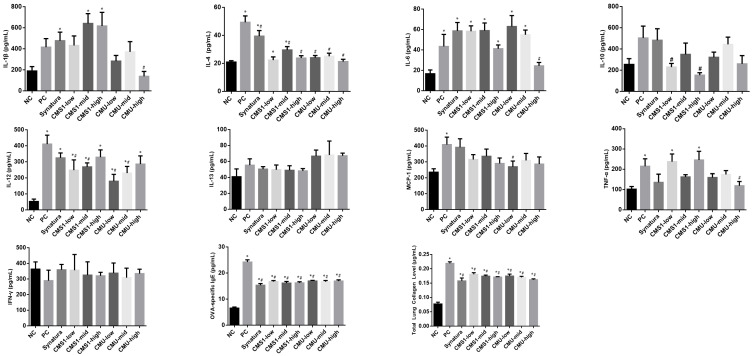
Effects of *W. cibaria* on cytokine, chemokine, and total lung collagen levels in BALB/c mice model of asthma induced by ovalbumin (OVA) plus diesel exhaust particulate matter (DEPM). The levels of cytokines in bronchoalveolar lavage fluid were measured by enzyme-linked immunosorbent assay. Synatura was used as an active control. Data represent the mean ± standard deviation. NC, negative control; PC, positive control; IL, interleukin; MCP, monocyte chemotactic protein; TNF, tumor necrosis factor; IFN, interferon; IgE, immunoglobulin E. * indicate the statistical differences with negative control (*p* < 0.05). # indicate the statistical differences with positive control (*p* < 0.05).

**Figure 4 medicina-58-01310-f004:**
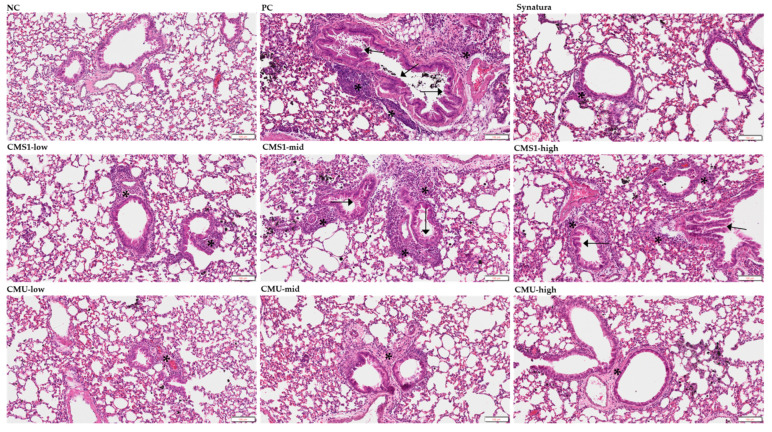
Effects of *W. cibaria* on histopathological changes in BALB/c mice model of asthma induced by ovalbumin (OVA) plus diesel exhaust particulate matter (DEPM). Lung tissues were collected, fixed in 10% neutral-buffered formalin, and stained with hematoxylin and eosin. Arrows indicate mucosal hyperplasia, and asterisks indicate inflammatory cell infiltration. NC, negative control; PC, positive control. Scale bars = 100 μm.

**Figure 5 medicina-58-01310-f005:**
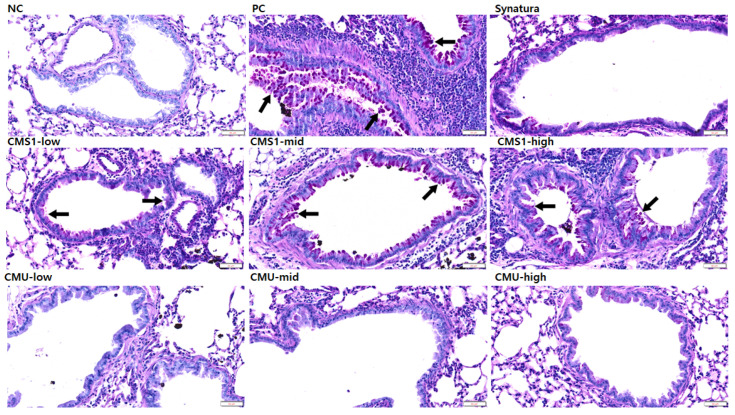
Effects of *W. cibaria* on mucosal hyperplasia in BALB/c mice model of asthma induced by ovalbumin (OVA) and diesel exhaust particulate matter (DEPM). The lung tissues were collected and fixed in 10% neutral-buffered formalin. The paraffin-embedded lung sections were stained with periodic acid–Schiff (PAS). Arrows indicate mucosal hyperplasia. NC, negative control; PC, positive control. Scale bars = 50 μm.

**Figure 6 medicina-58-01310-f006:**
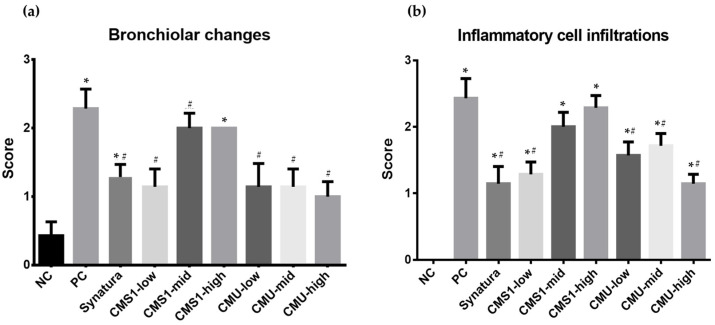
Effects of *W. cibaria* on bronchoalveolar changes (**a**) and inflammatory cell infiltrations (**b**) in BALB/c mice model of asthma induced by ovalbumin (OVA) and diesel exhaust particulate matter (DEPM). Data represent the mean ± standard deviation. NC, negative control; PC, positive control. * indicate the statistical differences with negative control (*p* < 0.05). # indicate the statistical differences with positive control.

**Table 1 medicina-58-01310-t001:** Scoring criteria for pulmonary histopathological examination.

Bronchiolar Changes		Inflammatory Changes	
Grade	Criteria	Grade	Criteria
0	Normal	0	Normal
1	Mild changes in bronchiolar epithelium, including mucosal hyperplasia, epithelial detachment, and bronchiolar smooth muscle hypertrophy	1	Mild infiltration of inflammatory cells, including interstitial lymphocytes, alveolar macrophages, and occasional eosinophils
2	Moderate bronchiolar changes	2	Moderate infiltrations of inflammatory cells
3	Severe bronchiolar changes	3	Severe infiltrations of inflammatory cells

## Data Availability

The data presented in this study are available upon request from the corresponding author.

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
