# Peer review of "Therapeutic Efficacy of Weissella cibaria CMU and CMS1 on Allergic Inflammation Exacerbated by Diesel Exhaust Particulate Matter in a Murine Asthma Model"

_medicina, 2022, doi:10.3390/medicina58091310_

Round 1
Reviewer 1 Report
Manuscript ID: Medicina-1859135
Thank you for the opportunity to review this manuscript entitled: “Therapeutic efficacy of Weissella cibaria CMU and CMS1 on Allergic Inflammation Exacerbated by Diesel Exhaust Particulate Matter in a Murine Asthma Model”.
This is an important topic demonstrating the safety and potential benefit of probiotic strains Weissella cibaria CMU and CMS1 in alleviating allergic inflammation.
· Overall, the manuscript is well written
· Consider including a sentence on how euthanasia of the mice was performed?
Author Response
<Comments from Reviewer 1>
Thank you for the opportunity to review this manuscript entitled: “Therapeutic efficacy of Weissella cibaria CMU and CMS1 on Allergic Inflammation Exacerbated by Diesel Exhaust Particulate Matter in a Murine Asthma Model”. This is an important topic demonstrating the safety and potential benefit of probiotic strains Weissella cibaria CMU and CMS1 in alleviating allergic inflammation.
Overall, the manuscript is well written
Consider including a sentence on how euthanasia of the mice was performed?
Response: I appreciate the time and effort that you and the reviewers have dedicated to providing your valuable feedback on my manuscript. According to your suggestion, I revised how euthanasia of the mice was performed on line 126..
Line 126: After the mice were sacrificed by over bleeding, bronchoalveolar lavage fluid (BALF) was immediately collected.

Reviewer 2 Report
In the manuscript “Therapeutic efficacy of Weissella cibaria CMU and CMS1 on Allergic Inflammation Exacerbated by Diesel Exhaust Particulate Matter in a Murine Asthma Model” Kyung-Hyo Do and collogues assess whether probiotic Weissella cibaria strain CMU and CMS1 can be used as a therapy to alleviate allergic asthma. To examine the therapeutic efficacy authors used well characterized 21 day allergic asthma model and challenge with Diesel Exhaust Particulate Matter. They reported, improved histopathological changes with data showing improved bronchiolar and inflammatory score and decreasing the levels of most Th2 cytokines in BALF in both Weissella cibaria strain CMU and CMS1 with strain CMU having better efficacy.
Strengths:
Good experimental design
Detailed methods
Weakness
Poor data representation
Confusing statistical significance representation
Poor quality figures
1. Statistical representation: This is a major concerns, throughout the manuscript in the figures and Table statistical significance is reported with superscript letters “a,b,c,d, abcd, bcd, cd, … etc” however by looking at the figure it is not clear which group is being compared with what? Also, it is not described in the methods or figure legends that what each superscript letter mean, other than that they are all p<0.05. This can be addressed by following standard reporting method for p values using * p<0.05, ** p<0.01, *** P<0.001, and a horizontal line should be added above the two groups being compared.
2. Figure 2, 3, 4, and Table 1 : it is not clear what is being compared and what is significant?
3. Table 1. and Figure 4: Authors mentioned that inflammatory cells including, interstitial lymphocytes, alveolar macrophages, and occasional eosinophils were examined. How these cells were identified. immunostaining for specific cell types (eosinophils, mast cells, neutrophils, lymphocytes, monocytes) should be done to support these results.
4. Line 206-210: better resolution images with higher magnification should be provided to support the results of infiltrating cells infiltrate.
5. Figure 3 and 4, are the quantitative scoring of histopathological analysis. If this data were obtained from Figure 5, then figure 5 should be presented first and figure 3 and 4 should be included as individual figure panels of quantitative measure instead of separate figures.
6. Table 2: This is an interesting data. It might be better presented this as each cytokine figure panel for a full figure and include the table as a supplementary data. The text describing these results be difficult to follow through because of the reported numbers and p values for each comparison in the text. This section should be revised to make it simple.
7. The cytokine levels were examined in the BALF, does BALF cell infiltrate was examined? Total cell count data in the BALF, and representative cytospine slides would add value to the manuscript.
8. In this study, OVA and DEPM challenge had no significant effect on the liver, spleen, or lungs weights, this is contrary to a previous study recently published by Jin et al. (2020), Authors did not provide an explanation on this observation.
9. Line 334-341: redundancy in discussion should be removed.
Author Response
<Comments from Reviewer 2>
In the manuscript “Therapeutic efficacy of Weissella cibaria CMU and CMS1 on Allergic Inflammation Exacerbated by Diesel Exhaust Particulate Matter in a Murine Asthma Model” Kyung-Hyo Do and collogues assess whether probiotic Weissella cibaria strain CMU and CMS1 can be used as a therapy to alleviate allergic asthma. To examine the therapeutic efficacy authors used well characterized 21 day allergic asthma model and challenge with Diesel Exhaust Particulate Matter. They reported, improved histopathological changes with data showing improved bronchiolar and inflammatory score and decreasing the levels of most Th2 cytokines in BALF in both Weissella cibaria strain CMU and CMS1 with strain CMU having better efficacy.
Strengths:
Good experimental design
Detailed methods
Weakness
Poor data representation
Confusing statistical significance representation
Poor quality figures
Comment 1: Statistical representation: This is a major concerns, throughout the manuscript in the figures and Table statistical significance is reported with superscript letters “a,b,c,d, abcd, bcd, cd, … etc” however by looking at the figure it is not clear which group is being compared with what? Also, it is not described in the methods or figure legends that what each superscript letter mean, other than that they are all p<0.05. This can be addressed by following standard reporting method for p values using * p<0.05, ** p<0.01, *** P<0.001, and a horizontal line should be added above the two groups being compared.
Response: According to your suggestion, I revised the statistical representation according to your suggestion.
Comment 2: Figure 2, 3, 4, and Table 1 : it is not clear what is being compared and what is significant?
Response: Same as the Comment 1. According to your suggestion, I revised the statistical representation according to your suggestion.
Comment 3: Table 1. and Figure 4: Authors mentioned that inflammatory cells including, interstitial lymphocytes, alveolar macrophages, and occasional eosinophils were examined. How these cells were identified. immunostaining for specific cell types (eosinophils, mast cells, neutrophils, lymphocytes, monocytes) should be done to support these results.
Response: Thank you for this suggestion. However, in the case of this manuscript, it seems slightly out of scope because HE staining was sufficient for identifying the types of inflammatory cells at high magnification (200 to 400 times). And immunostaining was not done because it was not necessary to determine the expression of specific markers on the cells.
Comment 4: Line 206-210: better resolution images with higher magnification should be provided to support the results of infiltrating cells infiltrate.
Response: Thank you for pointing this out. For reader friendly, I'll give the better resolution images to editor of MDPI.
Comment 5: Figure 3 and 4, are the quantitative scoring of histopathological analysis. If this data were obtained from Figure 5, then figure 5 should be presented first and figure 3 and 4 should be included as individual figure panels of quantitative measure instead of separate figures.
Response: According to your suggestion, I presented the Figure 5 and 6 first than Figure 3 and 4. For reader friendly, Figure 3 and 4 were merged on Figure 5.
Comment 6: Table 2: This is an interesting data. It might be better presented this as each cytokine figure panel for a full figure and include the table as a supplementary data. The text describing these results be difficult to follow through because of the reported numbers and p values for each comparison in the text. This section should be revised to make it simple.
Response: Thank you for your suggestion. However, the Table 2 expresses too many kinds of cytokine for nine experimental groups. For a quick and helpful understanding to reader, I revised the statistical representation according to your suggestion.
Comment 7: The cytokine levels were examined in the BALF, does BALF cell infiltrate was examined? Total cell count data in the BALF, and representative cytospine slides would add value to the manuscript.
Response: Thank you for your valuable feedback. The BALF cell infiltration was not investigated in this study. I'll investigate this in a subsequent study.
Comment 8: In this study, OVA and DEPM challenge had no significant effect on the liver, spleen, or lungs weights, this is contrary to a previous study recently published by Jin et al. (2020), Authors did not provide an explanation on this observation.
Response: Thank you for pointing this out. I revised the manuscript line 277-279 according to your comment.
Comment 9: Line 334-341: redundancy in discussion should be removed.
Response: According to your suggestion, Line 334-341 was removed.

Round 2
Reviewer 2 Report
The revised manuscript is improved, and authors have satisfactory addressed a few of my comments.
I have found one concern regarding the animal ethics and have a couple of comment regarding the Figure 4 and Table 2.
I believe providing higher magnification images will add value to the manuscript and will be very helpful for the readers to follow through the conclusions drawn from the manuscript. In the present form the conclusion drawn does not supported by the data presented in figure 4.
Specific comments.
In the revised version of the manuscript -
Line 126: Authors mentioned that the mice were sacrificed “by over bleeding”
This is a concern of the ethical guidelines, was the procedure approved in the animal protocol and institutional animal care and use guidelines? If so, it should be mentioned in the methods.
Additionally, the statement should be changed _ “The mice were euthanized ….”
Comment for - Authors response to comment #3.
“Response: Thank you for this suggestion. However, in the case of this manuscript, it seems slightly out of scope because HE staining was sufficient for identifying the types of inflammatory cells at high magnification (200 to 400 times). And immunostaining was not done because it was not necessary to determine the expression of specific markers on the cells. “
This is very important data of the manuscript. In that authors should consider including higher magnification images (200 – 400 times) in the manuscript pointing the each cell type that they identify, may be as a supplementary data. This will add value to the manuscript and will be very helpful for the readers.
Comment 6
Response: Thank you for your suggestion. However, the Table 2 expresses too many kinds of cytokine for nine experimental groups. For a quick and helpful understanding to reader, I revised the statistical representation according to your suggestion.
To authors, this is the main strength of the manuscript. There are a couple of ways this data can be represented in figure. Please have a look at the attached file.
1. Make graph for each individual analyte that was examined and keep the y axis same (can be broken in 2 for low level expression). Then add a figure panel.
2. Make 4 groups for example (Cytokine, Chemokine and Ige and collagen) and then make a graph and show graphically. Add statistical significance. I put IL-1b and IL-4 just as an example.

Author Response
Thank you for giving me the opportunity to submit a revised draft of our manuscript.
We appreciate the time and effort that you have dedicated to providing your valuable feedback on our manuscript.
We have been able to incorporate changes to reflect most of the suggestions provided.
We have highlighted the changes within the manuscript.
Here is a point-by-point response to the comments.
In addition to the comments, all references have been re-checked.
We look forward to hearing from you in due time regarding our submission and to respond to any further questions and comments you may have.
<Comments from Reviewer 2>
The revised manuscript is improved, and authors have satisfactory addressed a few of my comments.
I have found one concern regarding the animal ethics and have a couple of comment regarding the Figure 4 and Table 2.
I believe providing higher magnification images will add value to the manuscript and will be very helpful for the readers to follow through the conclusions drawn from the manuscript. In the present form the conclusion drawn does not supported by the data presented in figure 4.
<Specific comments 1>
In the revised version of the manuscript -
Line 126: Authors mentioned that the mice were sacrificed “by over bleeding”
This is a concern of the ethical guidelines, was the procedure approved in the animal protocol and institutional animal care and use guidelines? If so, it should be mentioned in the methods.
Additionally, the statement should be changed _ “The mice were euthanized ….”
<Answer to specific comments 1>
Response: According to reviewer’s suggestion, line 126 was modified to “The mice were euthanized...”
line 126: After the anesthesia, mice were euthanized by over bleeding
<Specific comments 2>
Comment for - Authors response to comment #3.
“Response: Thank you for this suggestion. However, in the case of this manuscript, it seems slightly out of scope because HE staining was sufficient for identifying the types of inflammatory cells at high magnification (200 to 400 times). And immunostaining was not done because it was not necessary to determine the expression of specific markers on the cells. “
This is very important data of the manuscript. In that authors should consider including higher magnification images (200 – 400 times) in the manuscript pointing the each cell type that they identify, may be as a supplementary data. This will add value to the manuscript and will be very helpful for the readers.
<Answer to specific comments 2>
Response: Thanks for this valuable comment. The histopathological changes were evaluated by one of our authors Dr. Soochong Kim, the ‘American College of Veterinary Pathologist’ board-eligible pathologist, and the identification of inflammatory cells which include lymphocytes, eosinophils, and macrophages are routinely conducted in H&E images by the pathologist based on their morphological features.
As a reviewer has suggested, we have identified and labeled the inflammatory cells in the higher magnification image of CMS1-high and PC as a supplementary figure. We uploaded supplementary Figure 1, and, for reader friendly, macrophages, lymphocytes, and eosinophilic infiltration were indicated in supplementary Figure 1 by black arrows, open arrows, and arrow heads, respectively.
Supplementary Figure 1 legend: Identification of inflammatory cells in higher magnification image of (A) CMS1-high and (B) PC group. Black arrows indicate macrophages, open arrows indicate lymphocytes, and arrow heads indicate eosinophilic infiltration. Scale bars = 50 μm.
<Specific comments 3>
Response: Thank you for your suggestion. However, the Table 2 expresses too many kinds of cytokine for nine experimental groups. For a quick and helpful understanding to reader, I revised the statistical representation according to your suggestion.
To authors, this is the main strength of the manuscript. There are a couple of ways this data can be represented in figure. Please have a look at the attached file.
- Make graph for each individual analyte that was examined and keep the y axis same (can be broken in 2 for low level expression). Then add a figure panel.
- Make 4 groups for example (Cytokine, Chemokine and Ige and collagen) and then make a graph and show graphically. Add statistical significance. I put IL-1b and IL-4 just as an example.
<Answer to specific comments 3>
Response: Thanks for giving us an example charts. According to reviewer’s suggestion, we modified Table 2 to Figure 3. As your suggestion, we tried to create a graph for each analyte with same y axis however, it was hard to read because of the size of data (eg. IL-1b: 200 pg/mL ~ 600 pg/mL; IL-6: 20 pg/mL ~ 60 pg/mL). So, all cytokines, chemokines and total lung collagen levels were expressed as column charts individually, and inserted as Figure 3. We think this form of presentation will be helpful for quick understanding to readers.
line: 217-219: Figure 3. Effects of W. cibaria on cytokine, chemokine, and total lung collagen levels in BALB/c mice model of asthma induced by ovalbumin (OVA) plus diesel exhaust particulate matter (DEPM). The levels of cytokines in BALF were measured by ELISA assay. Synatura was used as an active control. Data represent the mean ± standard deviation. NC, negative control; PC, positive control. * indicate the statistical differences with negative control (p < 0.05). # indicate the statistical differences with positive control (p < 0.05).
